# Hyaluronic Acid Nanogels: A Promising Platform for Therapeutic and Theranostic Applications

**DOI:** 10.3390/pharmaceutics15122671

**Published:** 2023-11-25

**Authors:** Su Sundee Myint, Chavee Laomeephol, Sirikool Thamnium, Supakarn Chamni, Jittima Amie Luckanagul

**Affiliations:** 1Department of Pharmacognosy and Pharmaceutical Botany, Faculty of Pharmaceutical Sciences, Chulalongkorn University, Bangkok 10330, Thailand; susundee@gmail.com (S.S.M.); supakarn.c@pharm.chula.ac.th (S.C.); 2Pharmaceutical Sciences and Technology Program, Faculty of Pharmaceutical Sciences, Chulalongkorn University, Bangkok 10330, Thailand; 6273011533@student.chula.ac.th; 3Department of Pharmaceutics and Industrial Pharmacy, Faculty of Pharmaceutical Sciences, Chulalongkorn University, Bangkok 10330, Thailand; chavee.l@chula.ac.th; 4Center of Excellence in Biomaterial Engineering in Medical and Health, Chulalongkorn University, Bangkok 10330, Thailand; 5Natural Products and Nanoparticles Research Unit (NP2), Chulalongkorn University, Bangkok 10330, Thailand; 6Center of Excellence in Plant-Produced Pharmaceuticals, Chulalongkorn University, Bangkok 10330, Thailand

**Keywords:** hyaluronic acid nanogels, theranostics, controlled release, targeted delivery, nanomedicine

## Abstract

Hyaluronic acid (HA) nanogels are a versatile class of nanomaterials with specific properties, such as biocompatibility, hygroscopicity, and biodegradability. HA nanogels exhibit excellent colloidal stability and high encapsulation capacity, making them promising tools for a wide range of biomedical applications. HA nanogels can be fabricated using various methods, including polyelectrolyte complexation, self-assembly, and chemical crosslinking. The fabrication parameters can be tailored to control the physicochemical properties of HA nanogels, such as size, shape, surface charge, and porosity, enabling the rational design of HA nanogels for specific applications. Stimulus-responsive nanogels are a type of HA nanogels that can respond to external stimuli, such as pH, temperature, enzyme, and redox potential. This property allows the controlled release of encapsulated therapeutic agents in response to specific physiological conditions. HA nanogels can be engineered to encapsulate a variety of therapeutic agents, such as conventional drugs, genes, and proteins. They can then be delivered to target tissues with high efficiency. HA nanogels are still under development, but they have the potential to become powerful tools for a wide range of theranostic or solely therapeutic applications, including anticancer therapy, gene therapy, drug delivery, and bioimaging.

## 1. Brief Introduction of Hyaluronic Acid

Hyaluronic acid (HA) is a ubiquitous glycosaminoglycan polymer in the extracellular matrices of connective, epithelial, and nervous tissues [1,2,3,4]. It is a negatively charged linear chain of repeating N-acetyl glucosamine and D-glucuronic acid units linked by alternating β-1,3- and β-1,4-glycosidic bonds [5,6]. Hyaluronidases cleave the β-1,4-glycosidic bond, mediating HA degradation [6]. HA exhibits diverse biological functions, such as cell proliferation, differentiation, and cell motility. Moreover, HA serves as a crucial mediator in the interplay between cells and their extracellular environment, facilitating interactions with specific HA receptors that translate HA signals into a cascade of cellular responses [7].

The most prominent HA receptor, present in almost all cell types, is the transmembrane glycoprotein cluster of differentiation 44 (CD44), which mediates cell adhesion and migration, as well as lymphocyte activation [8]. Another major HA receptor, found mainly in smooth muscle cells and endothelial cells, is the receptor for HA-mediated motility (RHAMM) [9]. The HA-RHAMM interaction mediates several signal transduction events and cytoskeletal interactions involved in cell growth and migration [9].

Moreover, HA’s unique physicochemical properties, namely viscoelasticity, biocompatibility, biodegradability, and high water absorption and retention capacity, make it a versatile lubricant, filler, and structural support in the human body [6]. HA’s molecular weight is a crucial parameter that determines its physicochemical and degradable properties. For example, increasing HA’s molecular weight can increase its viscoelasticity [10]. High molecular weight HA inhibits pro-inflammatory mediators, while low molecular weight HA induces cell proliferation and improves T cell and dendritic cell activation [11,12].

Additionally, HA can be conjugated and chemically modified to functionalize its properties for specific applications. The presence of a carboxyl group in D-glucuronic acid facilitates HA functionalization [1]. Covalent conjugation with amine groups is possible through amide bond formation, while covalent conjugation with hydroxyl groups is possible through esterification [13]. The carbodiimide reaction, a cross-linking reaction, can activate HA’s carboxyl groups for amide bond formation [1].

## 2. Hyaluronic Acid Nanogel

Nanogels, also known as nanoscale hydrogels, are typically spherical in structure and range in size from 1 to 1000 nanometers [5,14]. They are formed by chemical or physical cross-linking of a variety of synthetic or natural polymers, or a combination of hydrophilic or amphipathic polymers. Nanogels possess several distinct characteristics, including excellent colloidal stability, tunable swelling capacity due to their adjustable size, and high encapsulation capacity [5].

Nanogels are predominantly hydrophilic and swell in aqueous environments due to solvent penetration into their free spaces, resulting in volume changes [15]. Furthermore, nanogels possess a large surface area, which facilitates bioconjugation, functionalization, and the encapsulation of drugs and biological macromolecules within their network [5].

Additionally, nanogels can be self-assembled to achieve desired functionality. For example, stimulus-responsive nanogels can be prepared by conjugating stimulus-responsive moieties (molecules that respond to external stimuli by changing their properties) to their functional groups [1]. Depending on the type of external stimuli, such as physical stimuli (temperature, light, electromagnetic), chemical stimuli (pH, ionic strength, redox), or biological stimuli (glucose gradient, enzymes), nanogels respond by changing their shape, surface characteristics, solubility, and light transmission capacity [16].

### 2.1. Fabrication of Hyaluronic Acid Nanogels

Several approaches have been developed to prepare hyaluronic acid (HA)-based nanogels, as illustrated and summarized in Figure 1 and Table 1, respectively. Colloidal HA particles can be formed via electrostatic interaction with amine-rich cationic polymers, such as chitosan [17], polyethyleneimine (PEI) [18,19] or poly(β-aminoester) (PBAE) [20]. Anionic drugs or genetic materials can first form counterion complexes with the cationic polymers, which are then wrapped in an external particle structure via electrostatic interaction with the negatively charged HA. Alternatively, cationic molecules can be covalently linked to the HA backbone, and the modified HA can form polyplex nanogels with anionic cargos [21,22,23].

In addition to cationic polymers, metal ions can also be used as crosslinkers to form HA nanogels by coordinating with the functional groups on the HA backbone. For example, HA nanogels were physically formed by coating positively charged 3-thiol-1,2,4-triazole-based copper complexes [24]. 3-Thiol-1,2,4-triazole is an organic linker with triazole groups that function as bidentate ligands for Cu^2+^ and thiol groups that can be cleaved in the high reducing environment inside tumor cells. This metal–organic complex is a unique material called a nanoscale metal–organic framework (NMOF), which is applicable as a bioimaging agent [25].

Additionally, HA can be covalently conjugated with metal-chelating functional groups, such as histidine amino acids and chelating agents such as iminodiacetic acid and malonic acid. The chelating groups can form metal–ligand coordination with platinum-based chemotherapeutic drugs, such as cisplatin [26,27]. One advantage of physical complexation is its pH responsiveness. At acidic pH, the nanogel breaks down due to the ionization–deionization switch of the ligands, enabling the release of the payload at specific sites.

Amphipathic HA can also be obtained by conjugating hydrophobic moieties to the linear HA chain. The self-assembly of amphipathic HA into nanogels is driven by hydrophobic interactions, resulting in polymeric micelles in water. A variety of hydrophobic molecules have been trialed for this purpose, including linear alkyl chains [28,29,30,31,32,33,34], bulky hydrophobic molecules (e.g., cholesterol [35,36,37], pyrene [38,39], cholanic acid [40,41,42], and indocyanine green [43]), and hydrophobic polymers (e.g., ethylene glycol [44,45], polycaprolactone or PCL [46,47], poly(lactic-co-glycolic acid) or PLGA [48], and poly(*N*-isopropylacrylamide) or pNIPAM [49].

Hydrophobicized HA can spontaneously form micellar structures, with the hydrophobic region agglomerated and wrapped with hydrophilic HA to minimize surface energy with the aqueous interface. This structure is typically known as a polymerosome. The hydrophobic core of these nanogels can accommodate and protect hydrophobic cargos from hydrolysis, and control their release [50]. In addition to small hydrophobic molecules, proteins and peptides can also be entrapped in hydrophobic-modified HA nanogels via hydrophobic interaction. This method can preserve their folding structure, functions, and stability in labile environments [36,37].

A major limitation of spontaneously formed HA nanogels through hydrophobic assembly is their micellar-like nature, which is characterized by the presence of a critical aggregation concentration (CAC), or the lowest polymer concentration required to form the nanogel structure in aqueous systems. This feature is concerning when nanogels are intended for intravenous administration, as they may lose their supramolecular structure in the diluted circulatory fluid. Additionally, the hydrophobic interaction may be altered by serum proteins, remodeling the nanogel assembly in vivo [43,51]. Further chemical crosslinking of self-assembled nanogels helps to preserve the nanogel structure even at low concentrations [31] and prevent the premature release of payloads before reaching the target sites [43].

In addition to self-assembled nanogels formed through polyelectrolyte complexation or hydrophobic agglomeration, nanogels can be chemically stabilized using crosslinkers or modified HA with covalent-linkable groups. One of the most popular crosslinking reactions is the carbodiimide reaction, which catalyzes the formation of amide bonds between carboxyl and amine groups. This reaction is well known for its mild conditions (neutral pH, no organic solvents required) and ease of excess reactant removal. Since the D-glucuronic acid monomer of the HA backbone only contains a carboxyl group, amine linkers (e.g., ethylenediamine [52], lysine ester [53], or amino-containing polymers [5,54] are required to complete the crosslinking reaction. Additionally, crosslinking reactions between amino groups modified on the HA backbone have been achieved using aldehyde crosslinkers, such as terephthalaldehyde [55,56]. Other crosslinking agents include 4-(4,6-dimethoxy-1,3,5-triazin-2-yl)-4-methylmorpholinium chloride (a triazine derivative that can activate the carboxyl group of HA to form amide or ester linkages with amine or hydroxyl groups, respectively) [57], as well as divinyl sulfone and glycidyl ether, which can bridge the hydroxyl groups of N-acetyl glucosamine residues [5,58].

Nanogel crosslinking can exploit the disulfide bridge formation of thiolated HA. Cysteamine, a decarboxylated cysteine amino acid derivative, is a popular thiol-bearing molecule that is commonly conjugated to the HA backbone via amide formation. Crosslinked disulfide bonds can spontaneously form by simply agitating the thiolated HA solution [59,60] or by using an emulsion method to control the nanogel size through the size of the emulsified droplets [61,62,63].

The introduction of thiol groups to the HA chain is not limited to cysteamine. For example, thiol acid, such as thiopropionic acid, has been covalently linked to the hydroxyl group of the *N*-acetyl-*D*-glucosamine subunit of HA, and the thiol groups further form disulfide bonds, facilitating nanogel formation [64]. Long thiolated alkyl chains have also been grafted onto the HA backbone via amidation, and the nanogels readily form via hydrophobic interaction. Interestingly, the disulfide bridge of this thiol-modified HA does not form unless the thiol crosslinker 1,4-bis(3-[2-pyridyldithio]propionamido)butane (DPDPB) is applied [31,65].

Functionalization of the covalently linkable functional groups on the HA backbone is another approach in crosslinking the nanogel structure of HA. Methacrylate group is a feasible functional group that can be simply crosslinked. For example, the methacrylated HA nanogels were formed via a vinyl addition polymerization using acrylate or methacrylate derivatives, e.g., di(ethylene glycol) diacrylate or DEGDA [66,67], a newly synthesized methacrylate linker, dimethacroyloxy-ethoxypropane or DMAEP [68], bis-acrylamide derivatives [69,70], or polymers containing acrylate or methacrylate groups [71,72]. Indeed, the methacrylate group can be photopolymerized using UV or visible light with a presence of photoinitiators [73], or using a photo-clickable tetrazole-ene crosslinking reaction with a tetrazole-modified HA [74,75,76,77].

A major obstacle in fabricating nanogels using the chemical crosslinking method is controlling the morphology and size of the particles. HA can form either a large hydrogel network or a micron-sized particle. Crosslinking HA in a diluted solution can minimize the entanglement of the complex chains, allowing for full dispersion of the HA chains. Once the crosslinking reaction is complete, the discrete nanoparticles are formed [57]. Another reliable method is to disperse the HA solution in the aqueous phase of a water-in-oil microemulsion system before adding crosslinkers [5,73,78]. The size of the HA droplets is controllable through parameters such as the water-to-oil ratio, the surfactants used, and the force of mechanical agitation. The nanoparticle structure of HA can temporarily form, and the nanogels are then stabilized with chemical crosslinkers. For example, fluorescently labeled amine-modified HA was dissolved in a concentrated salt solution, and a bifunctional N-hydroxysuccinimide or NHS ester was used to form links between the amino groups [43]. Similarly, nanogels are transiently formed within a liposomal template, from which the hydrophilic HA localizes and forms a nanospherical structure in the aqueous core of the liposomes. The template is then removed after the crosslinking reaction [53].

**Table 1 pharmaceutics-15-02671-t001:** Hyaluronic acid nanogel fabrication techniques.

Fabrication Approach	Description	Advantages (+) and Limitations (−)	Refs.
Electrostatic interactions or polyelectrolyte complexation	Counterion formations with polycations, e.g., chitosan, polyethyleneimine (PEI) or poly(β-aminoester) (PBAE)	(+) Simple (+) Anionic small molecule drugs or macromolecule therapeutics entrapment (+) pH responsiveness	[17,18,19,20]
	Complex formation with metal cations	(+) Retrieve the unique characteristics of the entrapped metal cations, such as temperature activity or photoactivity (+) pH responsiveness	[24,25,26,27]
Self-assembly of amphipathic HA	Linear alkyl chains	(+) Effortlessly form nano-spherical structure (+) Entrap hydrophobic molecules and control the release (−) Loss of micellar structures in diluted conditions (−) Alteration on hydrophobic interactions from serum proteins	[28,29,30,31,32,33,34]
	Cholesterol		[35,36,37]
	Pyrene		[38,39]
	Cholanic acid		[40,41,42]
	Indocyanine green		[43]
	Ethylene glycol		[44,45]
	Polycaprolactone (PCL)		[46,47]
	Poly(lactic-co-glycolic acid) (PLGA)		[48]
	Poly(N-isopropylacrylamide) (pNIPAM)		[1,49,79,80]
Chemical crosslinking	Amide bond formation with amine linkers using carbodiimide reaction	(+) Robust nanogel structure (−) Difficult to control morphology and size of the particles (−) Extra steps to remove excess crosslinkers	[5,52,53,54]
	Crosslinking of amine-conjugated HA using aldehyde crosslinkers		[55,56]
	Specific crosslinkers for HA		[57,58]
Self-crosslinking	Disulfide formation	(+) Simple fabrication (+) Redox-responsiveness (for disulfide-formed HA nanogels)	[31,59,60,61,62,63,64,65]
	Self-crosslinking of methacrylated HA		[66,67,68,69,70]
	Photocrosslinking of methacrylated HA or tetrazole-modified HA 73–77		[73,74,75,76,77]

### 2.2. Stimulus-Responsive Properties of Hyaluronic Acid Nanogels

Stimulus-responsive materials exhibit the remarkable ability to alter their physical or chemical properties in response to specific stimuli. These stimuli can encompass a wide range of factors, including pH, light, temperature, redox potential, enzymes, external forces such as magnetic or electric fields, and even radiation. This unique property holds promise in the field of targeted drug delivery, as it allows for precise control of drug release at the desired site of action. By tailoring the material’s responsiveness to the specific conditions of the target site or by applying external stimuli, the release of encapsulated therapeutic agents can be regulated, maximizing efficacy while minimizing off-target effects [81].

A comprehensive overview of stimulus-responsive materials and their applications in the biomedical realm is provided by Shymborska et al. [82] and Alejo et al. [83]. These reviews offer valuable insights into the design and development of stimulus-responsive drug delivery systems, highlighting their potential to revolutionize therapeutic strategies.

Hyaluronidase-responsive HA nanogels are feasible as enzyme-responsive delivery systems, as their enzymatic degradation into small fragments can dictate the release of encapsulated cargoes. Hyaluronidases are widely distributed in various organs and extracellular fluids [84] and their expression is upregulated during inflammation, such as cancer metastasis, which involves extracellular matrix remodeling, vascularization, and cell migration [85,86]. The enzyme-triggered release approach can prevent premature payload leakage before reaching target cells that overexpress hyaluronidases [42,87]. Some encapsulated photoluminescent probes are typically quenched while entrapped within the HA structure, but their photoluminescence properties can be restored upon cleavage of the HA chains. This feature enables the development of traceable tools for imaging cells that overexpress hyaluronidases, such as cancer cells, by determining the luminescence intensity of specific tissues [43].

The pK_a_ of the carboxyl groups of D-glucuronic acid subunits is approximately 3–4, indicating that they are fully ionized at physiological pH (around 7) [88]. This anionic character allows HA nanogels to encapsulate cationic molecules, such as weak-base drugs (e.g., doxorubicin) or amine-rich polymers (e.g., PEI, PBAE, or chitosan) via polyelectrolyte complexation [18,87]. Doxorubicin, a chemotherapeutic agent with a pK_a_ of ~8 for its amine groups, can be loaded into HA nanogels through electrostatic interactions. Release can be triggered within tumor cells, which have a slightly acidic cytosolic pH compared to normal cells [89]. This acidic environment results in increased protonation of the carboxyl groups of HA, which neutralizes its overall charge and leads to doxorubicin decomplexation [54,61,69].

The pH-responsive release of drugs from HA nanogels is also observed in systems formed by metal–ligand coordination. For example, an increase in the release of Mn^2+^, a magnetic resonance imaging (MRI) probe, was observed from histidine-grafted HA nanogels at pH 5.5 compared to pH 7.4, due to the pH responsiveness of the metal–histidine coordination bond and the protonation of the carboxyl groups of HA [26]. Additionally, acid-cleavable linkages can be introduced into HA nanogels using several crosslinkers, such as 2,2-dimethacryoloxy-1-ethoxypropane (DMAEP) [68], ester bond-bearing crosslinkers, such as di(ethylene glycol) diacrylate or DEGDA [66], and ester-modified pluronic L61 linker [71]. The cleavage of these acid-labile linkages facilitates nanogel breakdown specifically in tumor cells, where the acidic tumor microenvironment can be exploited for targeted drug release.

Temperature-sensitive HA nanogels can be obtained by grafting thermoresponsive polymers onto the HA chains, such as di(ethylene glycol)methacrylate (DEGMA)-based polymers [44,90], or pNIPAM [49]. These polymers are hydrophobic relative to HA, resulting in self-assembly into nanogel structures driven by hydrophobic interactions. At temperatures above their lower critical solution temperature (LCST) (LCST of DEGMA ~34 °C, LCST of pNIPAM ~32 °C), these polymers become more hydrophobic, resulting in enhanced physical crosslinking between the hydrophobic segments and the accommodation of hydrophobic or poorly water-soluble molecules within the nanogels. The limited access of water molecules into the hydrophobic core of the nanogels can enhance the stability of water-labile molecules [79]. This temperature-dependent behavior can be used to control the release of payloads from the nanogels. Burst release can be avoided, and sustained release controlled by diffusion can be achieved [81,91].

Redox-responsive delivery systems are another approach to targeting cancer treatment. Glutathione (GSH), a biological reducing agent, is present in tumor cells at a significantly higher concentration (up to 10 mM) than in normal cells and extracellular fluids (approximately 2 to 20 µM) [92,93]. This approach enables relatively high drug localization in tumor cells, enhancing chemotherapeutic efficacy and reducing unwanted side effects in normal cells. Disulfide bonds are redox-responsive covalent linkages that can be cleaved in a highly reducing environment. HA nanogels crosslinked with disulfides can prevent the leakage of loaded cargo until the nanogels are internalized by tumor cells [59,61,75]. In addition to triggering the release of payloads in response to high GSH levels, disulfide cleavage can also be used as a switch for luminescent cargos. The light conversion properties of these materials are typically quenched while entrapped in nanogels but are recovered once the nanogels break down [69]. This feature can be applied in the development of in vivo bioimaging probes with sensor properties specific to tumor tissues.

### 2.3. Functions of Hyaluronic Acid Nanogels

Herein, the functions of the HA nanogel are summarized and depicted in Figure 2. HA nanogels, similar to other nanoencapsulation delivery systems, can improve the colloidal stability of payloads. Inorganic nanomaterials and cationic molecules can potentially aggregate in aqueous buffers due to electrostatic interactions among themselves or with serum proteins [26,59,94]. These aggregates can trigger host immune responses through complement activation when administered intravenously [95]. HA nanogels leverage their anionic nature to generate repulsive forces, which results in improved particle dispersion and prevents payload aggregation, precipitation, and nonspecific interactions with serum proteins. This feature can dictate the biodistribution of loaded cargoes in vivo by extending retention in the circulatory system and limiting nonspecific distribution and early elimination by immune cascades [65]. In fact, the systemic distribution of HA nanogels is similar to that of PEGylated nanoparticles, a method for nanoparticle coating using hydrophilic polyethylene glycol (PEG). Quattal et al. [96] and Aldawsari et al. [22] showed that PEG-based nanogels were retained in the circulatory system longer than HA-based nanogels, resulting in greater drug accumulation at target sites (e.g., tumors). This is because systemic retention time is a dominant factor influencing drug accumulation in tumor tissues, rather than the tumor targetability of HA [96]. The presence of PEG in HA-based nanogels was suggested to improve retention time in the circulation and achieve higher drug concentration at target sites [97].

The presence of HA enables lyophilization and reconstitution of nanogels without altering their physicochemical properties or potency of the payloads [98]. HA acts as a cryoprotectant, preserving the physical properties of polymeric nanoparticles [99] and liposomes [100,101] during freeze–thaw cycles and reconstitution. As a result, HA nanogels can be subjected to drying technologies, which improves their shelf life and facilitates product distribution.

One of the unique characteristics of HA nanogels is their tumor homing ability, which they achieve through passive and active targeting mechanisms. Like other nanodelivery systems, HA nanogels preferentially accumulate in tumor tissues due to the enhanced permeability and retention (EPR) effect [102]. This effect is caused by the leakage of the loosened tumor vasculature and the limited lymphatic drainage from tumors. Additionally, HA is a ligand for several cell surface receptors, including CD44, LYVE-1, RHAMM, and HARE [103]. These receptors mediate the internalization of HA nanogels through receptor-mediated endocytosis, tissue homing, and cellular signal transduction. Overexpression of CD44 receptors on tumor cells, which occurs during inflammation, allows HA nanogels to actively target tumor tissues [104]. Several studies have reported the significant accumulation of HA nanogels in tumor xenografts [26,40,59,75].

However, several studies by the group of Prof. Nicola Tirelli have shown that CD44 expression alone cannot fully explain the binding and internalization of HA-based materials, as high CD44 expression can also be found in normal cells. Other factors that can affect the binding and internalization of HA nanoparticles include hyperactivity of the receptors, CD44 mutations, and cell state (cancerous vs. normal cells) [105,106]. Notably, the high expression of CD44 variants in cancer cells has been associated with increased binding and internalization of HA-based nanomaterials [106]. In fact, high CD44 expression with high binding and uptake activities is not limited to cancer cells. Antigen-presenting cells, such as dendritic cells and macrophages, express a significant amount of CD44 receptors to regulate their immunological functions, especially during inflammatory conditions [107]. Therefore, HA nanogels can be designed to specifically target macrophages or dendritic cells in vivo, or even used in adoptive cell therapy approaches [62,108,109].

The biodistribution of HA nanogels in tumor xenograft models revealed that they accumulate not only at the grafted tumor sites but also in clearance organs, such as the liver and spleen, and high perfusion organs, such as the heart, lungs, and skin [65]. This is because nanomaterials of a specific size are eliminated by the reticuloendothelial system (RES), which is comprised of phagocytic cells that reside in the liver and spleen [110]. Additionally, the high expression of HARE and LYVE-1 on hepatic cells may contribute to the early distribution of HA nanogels to the liver [40,75]. As mentioned above, the presence of PEG in HA nanogel formulations can facilitate immune escape, extending retention time in the circulatory system.

Another notable feature of HA nanogels is their ability to quench encapsulated photoluminescent molecules. This means that the photoactivity of these payloads is low or undetectable unless the nanogels are internalized and destabilized or degraded by the stimuli they respond to, such as hyaluronidase enzymes or a highly reducing environment. For example, indocyanine green is an FDA-approved fluorescent probe for bioimaging. Its hydrophobic nature facilitates encapsulation within HA nanogels, and its fluorescence is quenched. When nanogels are internalized and ICG is released from the disassembled nanogels within the cytosol, the dequenching effect occurs [20,43]. This strategy has also been applied to pheophorbide, a photosensitizer that can generate reactive oxygen species upon light irradiation. Its photodynamic activity is inhibited while encapsulated in HA nanogels but is recovered when the nanogels are internalized and broken down within tumor cells [111].

## 3. Potential Therapeutic or Theranostic Applications of Hyaluronic Acid-Based Nanogels

HA-based nanogels hold promising features that make them potentially useful in a wide range of biomedical and theranostic research areas, including anticancer therapy, delivery systems for biomacromolecules such as genetic materials and proteins, and targeted delivery systems for immune cells. Theranostics is an emerging concept that describes a therapeutic approach combining diagnosis, such as imaging probes or real-time tracking systems, with a targeted treatment approach [112]. The potential applications of HA-based nanogels for both diagnosis and treatment are discussed and showcased in Figure 3.

### 3.1. Hyaluronic Acid-Based Nanogels as a Platform for Imaging-Guided Theranostic Anticancer Therapy

Due to their unique structure, which features a highly hydrated porous structure and a hydrophobic core to accommodate various types of molecules, HA nanogels can be loaded with multiple components to serve both diagnostic and therapeutic purposes. Bioimaging contrast agents, such as fluorescent dyes (e.g., cyanine, Alexa Flour [65], indocyanine green [20,43], BODIPY [97], and pyrene [39]), metal-based nanoparticles (e.g., gold nanomaterials [61,69], and iron oxide nanoparticles [64]), quantum dots or carbon dots [54], and medical contrast agents (e.g., iodixanol [75]), can be simply entrapped within the nanogels or covalently conjugated to the HA backbone before forming the nanogel structure. Chemotherapeutic agents, or photothermal or photodynamic agents, can be simultaneously loaded to achieve the desired therapeutic outcome. It is noteworthy that the localization of these encapsulated or grafted molecules within the nanogels is important, as their surface localization can modify the surface properties of the nanogels, which can affect biodistribution [65].

Hyaluronic acid (HA)-based nanogels are unique drug delivery vehicles that combine the intrinsic targetability of HA to CD44-overexpressing cancer cells with the stimulus-responsive behavior of the nanogels. Coumarin derivatives are near-infrared (NIR) and ultraviolet (UV)-responsive moieties that can convert light energy into heat. When HA-based nanogels are modified with coumarin derivatives, they exhibit photodynamic activity, whereby heat generation can facilitate the precise release of payloads within cancer cells [90]. Hang et al. developed coumarin-grafted HA nanogels that can target CD44 receptors and trigger intracellular drug release upon NIR or UV irradiation [113]. These nanogels are internalized by CD44-positive cells via receptor-mediated endocytosis. Upon NIR and UV irradiation, the nanogels swell and destabilize, leading to the intracellular release of the drug payload. In vitro studies have demonstrated that these nanogels exhibit significant antitumor activity against cancer cells, without negative effects on normal cells with low CD44 expression.

Multidrug resistance (MDR) in cancer is a major clinical challenge that limits the efficacy of chemotherapy [114]. To overcome this, researchers have explored the combined delivery of phototherapeutic and chemotherapeutic agents. Khatun et al. developed a redox-responsive HA nanogel that encapsulates light-responsive graphene and the chemotherapeutic drug doxorubicin [59]. Upon laser irradiation, the graphene converts the light energy into heat, which induces the release of doxorubicin from the nanogel. This combination of photothermal therapy and chemotherapy was shown to enhance the antitumor effect and reverse MDR in genetically modified cancer cells.

Pan et al. developed HA-based nanogels for imaging-guided chemo-photodynamic cancer therapy by encapsulating the photodynamic agent chlorin e6 and the chemotherapeutic drug doxorubicin, and crosslinking the nanogels with Mn^2+^, an MRI contrast agent [26]. The nanogels were responsive to acidic pH and high reducing conditions, resulting in the release of the payloads within tumor cells upon nanogel breakdown. Both in vitro and in vivo studies demonstrated that the nanogels were selectively delivered to CD44-overexpressing B16 melanoma cells, and that T_1_-weighted MRI could be used to monitor tumor progression and treatment response.

### 3.2. Hyaluronic Acid-Based Nanogels as a Nonviral Vector for Targeted Intracellular Gene Delivery

The administration of naked genetic materials, such as plasmid DNA, oligonucleotides, short-stranded RNA, and messenger RNA (mRNA), is impossible without the use of viral or nonviral vectors. This is due to the rapid degradation of genetic materials by enzymes in the plasma and interstitial fluid, as well as the repulsion between the negatively charged genetic materials and the negatively charged cell membrane [115]. The complexation of genetic materials with cationic lipids (lipoplexes) or cationic polymers (polyplexes) can form nanosized complexes with a positive surface charge, which can facilitate cellular uptake. However, this approach can also lead to cell death because of disruption of the cell membrane. Shielding or polyelectrolytic complexing with anionic HA can reverse these negative effects, and HA can also mediate cellular uptake through receptor-mediated endocytosis [116].

PEI is an amine-rich polymer with a pK_a_ of approximately 8–9. At physiological pH, the amine groups of PEI are protonated, which gives PEI a positive charge. This positive charge allows PEI to form electrostatic complexes with negatively charged genetic materials. PEI polyplexes are widely studied gene delivery vectors due to their high and reliable transfection efficiency. However, PEI is nonbiodegradable, which limits its clinical and commercial applications. Researchers have investigated modified PEI derivatives that are biodegradable and have similar transfection efficiency to PEI [117,118,119].

Chitosan is a biodegradable, naturally derived cationic polymer that can form polyelectrolyte complexes with anionic genes. These complexes can be further shielded with HA to improve cytocompatibility and cellular uptake [106]. The physical and chemical properties of the chosen chitosan can affect the transfection efficiency of the formulations. For example, ternary complexes of mRNA, HA, and high molecular weight chitosan have high stability in plasma and a high internalization rate but low transfection efficiency due to the strong electrostatic interaction that prevents the mRNA release. In contrast, complexes formed with low molecular weight chitosan or chitosan with a low degree of acetylation are unstable and poorly taken up by cells [120]. Another study found that polyplexes formed using only an electrostatic interaction are susceptible to disruption due to the presence of oppositely charged molecules [121]. Larger genes, such as plasmid DNA, have a stronger affinity for polycations than smaller genes, such as mRNA and small interfering RNA. However, the physical electrostatic forces can be weakened by HA, depending on the amount and chain length of HA. At an optimal ratio of chitosan and HA, stable nanocomplexes with high transfection efficiency can be achieved.

To overcome the limitations of polyelectrolyte complexation, gene-loaded HA nanogels can be prepared using HA grafted with cationic polymers, such as HA-PEI [22,122], HA-spermine [94,123], HA-poly(2-(dimethylamino)ethyl methacrylate) or HA-pDMAEMA [51]. Covalent crosslinking of gene-loaded HA nanogels can achieve a stable nanogel structure, and the release of gene payloads can be triggered by cleavage of stimulus-responsive bonds, such as the disulfide crosslinking of thiolated HA [63,124]. Hydrophobicized HA can also facilitate self-assembly into the nanogel structure or form a complex with lipoplexes through hydrophobic interactions [36,125].

### 3.3. Hyaluronic Acid-Based Nanogels as Targeted Protein Delivery Vehicles

In recent years, the development of therapeutic proteins as alternative drugs to conventional drugs has gained widespread interests, especially for the treatment of cancer and other diseases. However, the systemic delivery of therapeutic proteins is a challenge due to their low bioavailability, short in vivo half-life, and physical and chemical instability [37]. To address the high unmet medical needs in protein delivery, many strategies have been adopted, one of the most promising of which is the use of HA-based nanogels. HA-based nanogels are potentially unique systems for protein loading and delivery due to their distinctive characteristics, including a hydrophilic shell that prevents protein degradation and an extensive internal space to encapsulate macromolecules.

Nakai et al. pioneered the use of cholesterol to form amphipathic HA nanogels for the encapsulation and sustained release of proteins, including human growth hormone recombinant, erythropoietin, exendin-4, and lysozyme. These nanogels demonstrated remarkable physical and chemical stability, preserving the bioactivity of the encapsulated proteins in vivo [37]. In a subsequent study, the same group demonstrated that cholesterol-bearing HA nanogels could also protect encapsulated antibodies from protein degradation due to thermal denaturation [126]. HA–cholesterol has also been applied to siRNA (small interference RNA, a short double-stranded RNA that can interfere with the translation of specific genes) delivery. HA–cholesterol was complexed with a virus 2b protein and siRNA, resulting in increased siRNA stability against enzymatic degradation and pH-dependent intracellular release [36].

Multifunctional HA nanogels are another potential theranostic option for targeted protein delivery. Chen et al. reported intrinsically fluorescent HA nanogels that encapsulate two intracellular protein drugs, cytochrome C and granzyme B [76]. The nanogels were constructed using the photoclick reaction of tetrazole-alkene, resulting in a more stable nanostructure than physically crosslinked HA nanogels. The delivery of cytochrome C and granzyme B was directed to overexpressed CD44 cancer cells, and the nanogels could be tracked by fluorescent emission. The photocrosslinked HA nanogels were further optimized for intracellular release by introducing a disulfide bond, which resulted in nanogel breakdown in the high reducing environment of tumor cells. The efficiency of protein delivery to tumors was improved, resulting in high tumor cytotoxicity and biodistribution to tumor cells, which could be visualized using the intrinsic fluorescent properties of the nanogels [77].

### 3.4. Hyaluronic Acid-Based Nanogels as a Promising Platform for Ocular Drug Delivery

HA is an abundant component of the extracellular matrix of the eye tissue and serves as a viscoelastic fluid with mucoadhesive properties [127,128]. Therapeutic agents encapsulated in HA-based nanogels retain their biological activity and allow for sustained release [128]. For example, Laradji et al. synthesized self-assembled, redox-responsive HA nanogels composed of thiol-bearing cystamine and hydrophobic cholesterol [129]. The nanogels were complexed with cell-penetrating peptide, penetratin, for targeted delivery to the posterior segment of the eye via topical application. The visual chromophore analog, 9-*cis*-retinal, was loaded into the HA-based nanogels, which were able to reach the retinal pigmented epithelium (RPE) cells and deliver their load in response to the reduced cellular environment.

In addition, the mucoadhesive properties of HA can mediate transcellular penetration when applied topically to the eye mucosa. Ternary complexes of HA, chitosan, and plasmid DNA have been shown to penetrate through epithelial junctions, enhancing the accumulation of the nanogels in cells [130,131]. Another challenge of ocular drug delivery is the presence of mucin, a highly viscous liquid that can hinder cell binding and internalization [131,132]. However, the targetability of HA to eye cells that overexpress CD44 receptors during inflammation, the anionic surface of HA nanogels, and their high retention in the vitreous fluid make them an effective delivery system for ocular targeting therapy.

### 3.5. Hyaluronic Acid-Based Nanogels for Skin Regeneration and Antimicrobial Therapy

HA exhibits a remarkable abundance in the skin dermis. HA-based formulations have been recognized for their diverse therapeutic and cosmetic applications, notably in facilitating wound healing, promoting tissue regeneration, and enhancing skin rejuvenation [133]. A comprehensive review of HA’s roles in dermatology has been recently published by Mauri and Scialla [134].

Several studies have explored the incorporation of HA into nanogel formulations for wound healing applications, leveraging HA’s inherent properties in skin regeneration. HA has been covalently conjugated onto various polymer backbones, including silk fibroin and polylactic acid, resulting in nanogels capable of encapsulating bioactive agents for dermatological applications, such as curcumin and verteporfin, respectively [34,135]. The presence of HA in the nanogel composition can augment the wound healing process and enhance skin penetration of the entrapped drugs to deeper layers.

Furthermore, HA-based nanogels hold promise for the delivery of antimicrobial agents. Fasiku et al. introduced the utilization of HA nanogels for the co-delivery of nitric oxide and an antimicrobial peptide against bacterial biofilms. This combination delivery exhibited broad-spectrum antibacterial activity against both Gram-positive and Gram-negative bacteria. Moreover, the presence of HA prolonged the half-life of the nitric oxide donor and imparted a suitable viscosity for topical administration [136]. Similarly, Kłodzińska et al. encapsulated a cytotoxic antimicrobial peptide within HA nanogels. Their findings demonstrated that the formulated peptide exhibited significantly reduced toxicity upon topical application while maintaining its effectiveness against pathogenic bacteria [137].

### 3.6. Hyaluronic Acid-Based Nanogels as Targeted Delivery Vehicles for Immune Cell Modulation

As mentioned above, immune cells express a certain amount of HA-specific receptors, such as CD44, especially during inflammation [107]. Therefore, HA nanogels have been investigated as a means to specifically target immune cells, particularly antigen-presenting cells, such as dendritic cells and macrophages. These cells typically overexpress CD44 receptors, and their phagocytic nature can facilitate the uptake of nanogels.

Targeted delivery to antigen presenting cells an facilitate the development of vaccine delivery systems. Kim et al. designed dual-targeting HA-based nanogels conjugated with ovalbumin as an antigen and decorated with schizophyllan, a β-glucan derivative, to target dendritic cells. Topical administration of the developed nanogels can induce the activation and maturation of dendritic cells [138].

Targeted delivery to macrophages, both circulatory and ex vivo adoptive cells, can be achieved using HA nanogels based on the CD44 targetability of HA. In a study by Tran et al., PEI-grafted HA nanogels were able to directly deliver plasmid DNA to macrophages after intravenous administration, thereby modulating their polarization [139]. Additionally, the tumor-homing capability of macrophages can be combined with nanotherapeutic systems. Xiao and colleagues loaded macrophage cell lines with HA nanogels containing the chemotherapeutic drug doxorubicin and polypyrrole, a photothermal agent that responds to NIR radiation. Compared to direct administration of nanogels, nanogel-entrapped macrophages demonstrated greater tumor homing and retention abilities, as evidenced by the increased tumor-to-liver accumulation ratio. Furthermore, encapsulation of doxorubicin in nanogels protected macrophages from cytotoxicity, and the drug could be released by thermal conversion of polypyrrole after NIR irradiation [62].

### 3.7. Hyaluronic Acid-Based Nanogels for MRI-Aided Theranostic Applications in Alzheimer’s Disease

Alzheimer’s disease is an irreversible and progressive neurodegenerative disorder caused by the abnormal accumulation, aggregation, and deposition of extracellular amyloid β-protein (Aβ) plaques in the brain [140]. While there is currently no cure for Alzheimer’s disease, inhibiting Aβ accumulation is a promising strategy to slow disease progression. Researchers have developed a variety of inhibitors to prevent Aβ plaque aggregation, including HA nanogel-based treatments. HA nanogels can readily bypass the blood–brain barrier with high efficiency, making them ideal for delivering Aβ aggregation inhibitors to the brain [64,140]. In a study by Jian et al., the synergistic effect of two hydrophobic inhibitors, epigallocatechin-3-gallate and curcumin, on Aβ aggregation was evaluated [140]. The incorporation of these two inhibitors into HA nanogels together with a favorable nanostructure exhibited enhanced Aβ aggregation inhibition.

Zhang et al. demonstrated the theranostic application of HA nanogels in Alzheimer’s disease by loading them with iron oxide nanoparticles as an MRI contrast agent [39]. Hyaluronidase in target cells caused the nanogels to break down, and the light intensity increased in response to the degradation of the nanogel. Chen et al. further validated these results in a similar study, in which iron oxide nanoparticles were fabricated and encapsulated in situ within HA nanogels using thiolated HA as a precursor [64]. The nanoparticle-entrapped HA nanogels exhibited a noticeable superparamagnetic property, which is favorable for MRI applications. Additionally, in vitro results showed the nanogels’ potential ability to inhibit Aβ aggregation, confirming the theranostic application of iron oxide nanoparticle-loaded HA nanogels in disease diagnosis and prevention of amyloid plaques.

## 4. Conclusions

HA nanogels have emerged as promising therapeutic carriers in the biomedical field owing to their unique combination of properties. Their ability to encapsulate and deliver therapeutic agents in a controlled manner, coupled with their stability enhancement and targetability, makes them attractive as an excellent drug delivery platform.

The inherent affinity of HA toward cell surface receptors, such as CD44 and RHAMM, allows HA nanogels to selectively target cancer cells that overexpress these receptors. This targeted approach maximizes therapeutic efficacy while minimizing off-target effects, offering a significant advantage over conventional therapies.

The presence of functional groups along the HA backbone facilitates simple and diverse fabrication methods for HA nanogels, including polyelectrolyte complexation, chemical crosslinking, and chemical modification. Despite some limitations, these methods still enable the production of self-assembling or self-crosslinked nanogels. The incorporation of self-responsive moieties into HA nanogels adds an extra layer of control, triggering drug release in response to specific microenvironmental cues or external stimuli. This feature enables the nanogels to deliver their cargo in a controllable manner.

HA-based nanogels hold promise for diverse biomedical applications due to their unique properties, including active targeting of cancer cells and stimulus-responsive drug release. They also offer an alternative delivery platform for gene and protein delivery, in vivo real-time diagnosis, and targeted drug delivery to specific tissues. However, ongoing clinical studies and further investigations are crucial and still needed to validate their safety and efficacy for a wide range of therapeutic purposes. Nevertheless, the successful development of HA-based nanogels has the potential to revolutionize the future of healthcare by providing novel and effective treatment options.

## Figures and Tables

**Figure 1 pharmaceutics-15-02671-f001:**
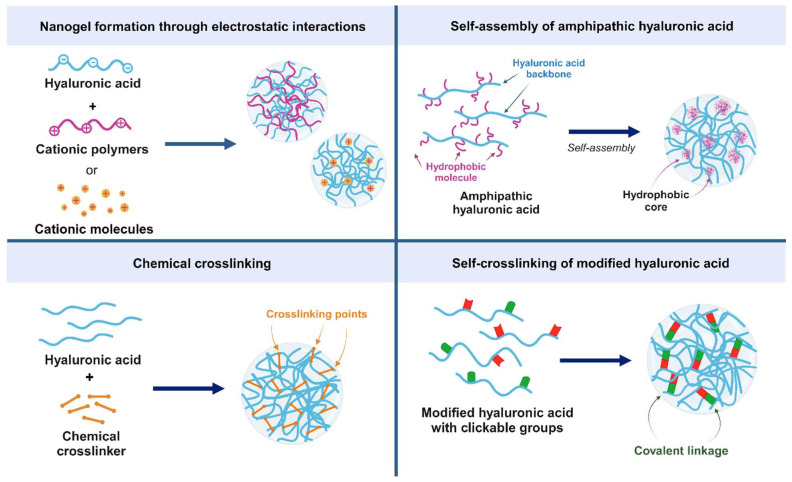
Strategies for fabricating HA nanogels. Hyaluronic acid (HA) nanogels can be fabricated using a variety of methods, including the following: (1) electrostatic interactions—cationic polymers or inorganic cations can be used to crosslink HA through electrostatic interactions; (2) self-assembly—amphipathic HA can self-assemble into nanogels driven by hydrophobic interactions; (3) chemical crosslinking—HA can be chemically crosslinked using different reactions, e.g., carbodiimide reactions, disulfide bond formation, or the crosslink reactions of methacrylate groups; (4) self-crosslinking—HA modified with covalently linkable groups can form nanogel structures in response to specific stimuli.

**Figure 2 pharmaceutics-15-02671-f002:**
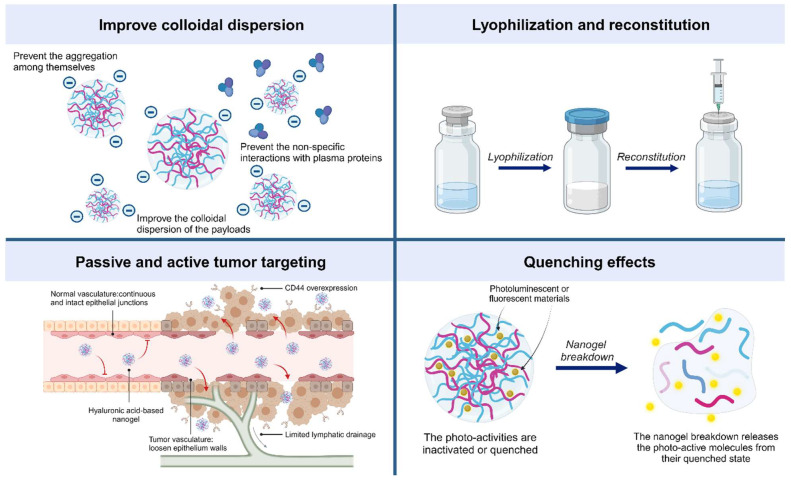
Functions of HA nanogels. Hyaluronic acid (HA) nanogels have several unique functions, including the following: (1) improved colloidal stability—HA nanogels can prevent aggregation and precipitation of encapsulated payloads, extending their retention in the circulatory system and limiting nonspecific distribution; (2) lyophilization and reconstitution—HA nanogels can be lyophilized and reconstituted without altering their physicochemical properties or potency of the payloads, improving their shelf life and facilitating product distribution; (3) tumor homing—HA nanogels can accumulate in tumor tissues through passive and active targeting mechanisms, including the enhanced permeability and retention (EPR) effect and receptor-mediated endocytosis; (4) quenching of encapsulated photoluminescent molecules—HA nanogels can quench the photoactivity of encapsulated molecules until they are internalized and destabilized or degraded; this can be used to control the release of payloads in response to specific stimuli or to improve the biocompatibility of photosensitizers.

**Figure 3 pharmaceutics-15-02671-f003:**
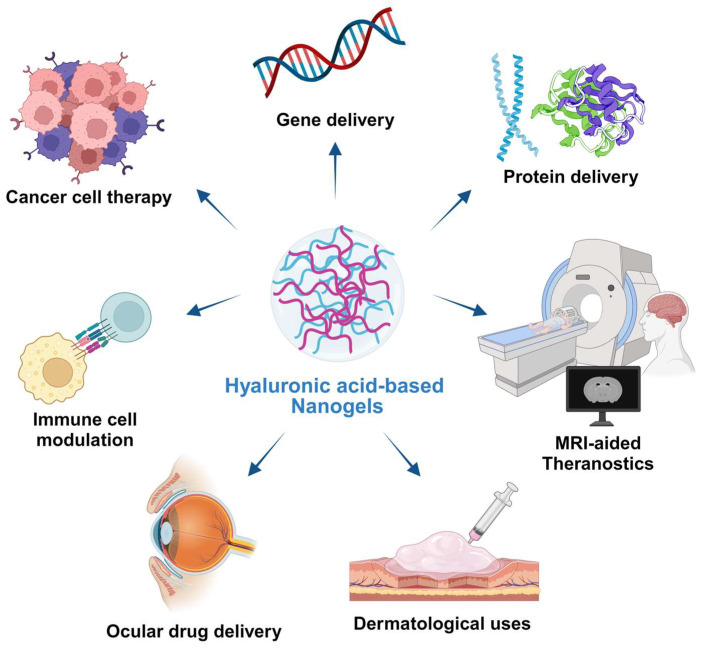
Potential applications of HA nanogels. Due to their unique properties, HA nanogels have emerged as promising candidates for theranostic or solely therapeutic applications. Their ability to encapsulate a diverse array of therapeutic agents, ranging from small molecule drugs to macromolecular therapeutics, coupled with their responsiveness to external stimuli for controlled release, makes HA nanogels ideal for a wide range of biomedical applications. HA nanogels have been extensively studied in various biomedical fields, including anticancer therapy, gene therapy, targeted drug delivery to specific tissue sites, and bioimaging.

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
