# Peer review of "Hyaluronic Acid Nanogels: A Promising Platform for Therapeutic and Theranostic Applications"

_pharmaceutics, 2023, doi:10.3390/pharmaceutics15122671_

Round 1

Reviewer 1 Report

Comments and Suggestions for Authors

This review manuscript focused on hyaluronic acid nanogels – their different types (including stimuli-responsive nanogels), methods of preparation, and potential applications. Overall, this manuscript has a lot of qualities, it is well designed and written, the text is comprehensive and generally well organized into sections. New and significant findings have been highlighted in the manuscript. Schematic presentations are also clear and make the text to be followed easier.

There are a few concerns/suggestions for the manuscript modification.

The main suggestion is to add one paragraph dedicated to skin drug delivery using hyaluronic acid nanogels. There is a recent review on this topic that might be useful (https://doi.org/10.3390/cosmetics10040113).

Another concern is regarding the terminology in the manuscript, specifically when mentioning “therapeutic and theranostic applications”. Since ‘theranostics’ is a term derived from a combination of words ‘therapeutics’ and ‘diagnostics’, it would be more appropriate to refer to “theranostic or solely therapeutic applications”.

Other concerns/suggestions are as follows:

Lines 24-28: It is not clear what this part is – highlights, part of the abstract or part of the introduction?

Line 39: It should be ‘conventional drugs’ instead of just ‘drugs’ since proteins can be drugs as well.

Line 53: It should be clear what kind of interaction the authors are referring to (cell-cell interactions?).

Line 71: Covalent conjugation with amide or amine groups?

Line 201: Define NHS ester.

Line 317: “cell stage (cancerous vs. normal cells)” is not completely clear.

Line 353: as previously said, the authors should refer to “theranostic or solely therapeutic applications” since theranostics includes therapeutics.

Line 458: Define siRNA.

Comments on the Quality of English Language

No major issues were detected.

Reviewer 2 Report

Comments and Suggestions for Authors

The review provides a concise summary of pertinent and intriguing findings. The fabrication, properties, and applications of nanogels are integral to modern chemistry. However, the review requires major revisions before it can be accepted, as the following points need clarification.

It would be valuable to present the results from section 2.1 in tabular format, including information on the fabrication method, chemical structure of the nanogel, specific properties, and references. This, in conjunction with Figure 1, can enhance the readability and structure for the readers.

One notable strength of the review is the inclusion of subsection "2.2. Stimuli-Responsive Properties of Hyaluronic Acid Nanogels." However, for non-specialized readers, it may be challenging to grasp the importance of this topic without additional context. I recommend incorporating and citing the following references (https://doi.org/10.1002/tcr.202300217 and https://doi.org/10.1016/j.jconrel.2019.10.036) to bolster the review's potential.

To engage the readers more effectively, it would be beneficial to illustrate Section "3. Potential Therapeutic and Theranostic Applications of Hyaluronic Acid-Based Nanogels" with figures showcasing successful examples of hyaluronic acid nanogels in biomedical applications.

The conclusion section requires substantial improvement. It currently lacks clarity regarding the advantages and disadvantages of hyaluronic acid-based nanogels when compared to analogs. Moreover, it would be beneficial to discuss the future trends in the development of hyaluronic acid-based nanogels.

Comments on the Quality of English Language

Minor editing of English language required.

Reviewer 3 Report

Comments and Suggestions for Authors

The manuscript is a critical review of the available data in the scientific literature. The review adresses important and relevant issues concerning with preparation, properties and applications of hyaluronic acid (HA) nanogels. The review consists of three parts, the first of which summarizes the properties of HA. The second part focuses on the methods of preparation HA nanogels and discusses stimuli-responsive nanogels responding to external influence such as pH, temperature, redox-potential etc. The uniqui functional properties of HA gells are discused in detail: colloidal stability, ability to lyophilization and reconstitution without loss of their physico-chemical properties, quenching of luminescence of encapsulated molecules, tumor homing. Of particular interest is the third section which reviews a wide range of applications of nanogels, particularly in anticancer therapy, targeted drug delivery, gene therapy, bioimaging etc. A large amount of information is analyzed in the review and the reference list includes 132 references. The review is written in good language and reads with interest. The review will be  usful for wide range of specialists working in the field of biomedicine and genetics

I recommend acceptance of this manuscript; however there is one comment that should be corrected:

Lines 292-293 and 333-334 - The word "fluorescence" should be deleted because fluorescence is a special case of luminescence.

Comments on the Quality of English Language

Minor editing of English language required

Round 2

Reviewer 1 Report

Comments and Suggestions for Authors

The authors accepted all the suggestions, so I support the manuscript to be published in this form.

Reviewer 2 Report

Comments and Suggestions for Authors

The authors have answered all my remarks and the review can be accepted in its present form.

Comments on the Quality of English Language

Minor editing of English language required.